# The Influence of Environment Factors on Chronic Non-Communicable Diseases in a Heavy Industry City— A Case of Xigu District of Lanzhou City

**Haili Zhao, Yuhan Du \*, Jialiang Li, Minghui Wu and Fang Zhang**

College of Geography and Environmental Science, Northwest Normal University, Lanzhou 730070, China; zhaohl@nwnu.edu.cn (H.Z.); 18851253591@163.com (J.L.); wmh17865575597@163.com (M.W.); zf20211101@163.com (F.Z.)
\* Correspondence: dyh2021nwnu@163.com

**Abstract:** Taking Xigu District of Lanzhou City as an example, this paper systematically analyzes the spatio-temporal distribution characteristics of patients with chronic non-communicable diseases (NCD) and compares the differences between heating period and non-heating period. Furthermore, the impact paths of natural environmental factors and built-up environmental factors on NCD are probed with the help of the geographic detector. The results are as follows: In time, the incidence of NCD in Xigu district fluctuated from 2012 to 2019. In space, there was an overall declining trend in high incidence rate from the central area to the surrounding areas, among which Xigucheng street was the high-risk area. The incidence of NCD in heating period was higher than that of in non-heating period, and the number of H-H cluster areas was witnessed an obviously increasing growth in Sijiqing Street. There are significant differences in the explanatory power of different factors (if any) for NCD. The explanatory power of each index in Xigu District is as follows: Facility > $SO_2$ > $NO_2$ > PM2.5 > food > Beverage Service > Green Facilities > Traffic Regulations > medical facilities. The interaction between plant facilities and $SO_2$ has the strongest effect on NCD. Except for the negative correlation between greening and medical facilities and the incidence of NCD, all the influencing factors were positively correlated with NCD.

**Keywords:** noninfectious chronic disease; heavy industry city; factors of environment; geographic detector; Xigu district



## 1. Introduction

"Chronic disease" is short for noninfectious chronic disease (NCD) which refers to sort of complex non-communicable disease caused by genetic and environmental factors as well as by lifestyle [1]. Since the WHO (World Health Organization) released the report of "Preventing Chronic Diseases: A Vital Investment" in 2006, chronic diseases have arisen more and more regard of countries all over the world [2]. Statistics show that 40 million people die of chronic diseases each year since the year of 2015. In China, there currently exist more than 200 million patients with hypertension and 90 million ones with diabetics. The fatality resulted from chronic ailments account for 75% of the overall death rate and has become one major bane for the loss of population [3,4]. In view of the detrimental impacts of chronic diseases to our people's health, *Healthy China 2030* Planproposed the goal: compared with 2015, China's chronic disease mortality rate to reduce 30% [5]. With people putting more focus on health issue, many fields ramp up investments on related research and what factors are running behind chronic diseases also has become a hot topic for scholars.

Scholars at home and abroad have long started to study the factors influencing NCD from a medical point of view and have verified the relationship between NCD and patients' dietary structure, age, marriage, culture, smoking and drinking habit, overweight and



obesity, sitting time and other personal behavior habits and individually social characteristics [6–11]. With the development of medical geography, the discipline has turned to study the distribution of diseases and health status among population and the influencing factors. Some scholars put an eye on the impact of the environment on NCD and have discovered that a link existed between the growth of NCD and patients' social factors, such as, occupation, family background, income in the economic environment; and green land area in the natural environment and air quality [12–17]. Domestic and foreign scholars make their study with the help of many methods such as the kernel density analysis [18], Moran index spatial autocorrelation [19], logistic regression model, ordinary least squares (OLS), and ordinary linear regression model, [20–24] etc. Existing research can be generally categorized as the one between NCD and environment (natural environment, built environment, social environment) [25–29]. In heavy industrial areas, compared to the social environment, the natural environment and built environment play a particularly important role in the health of urban residents, and they can easily induce diseases such as high blood pressure, heart trouble, and chronic obstructive pneumonia [28,29]. At the present stage, the influence path of natural and built environments on the health of urban residents urgently needs to be further revealed.

As far as the impact of the environment on NCD is concerned, scholars at home and abroad have carried out many empirical studies, but in the existing research, when exploring the impact mechanism of air pollution on NCD, they often use months and years as time scales to study its seasonal changes while in heavy industrial areas or other special regions, other time scales have not been considered yet [30–34]. Meanwhile, most studies only consider the impact of the natural environment or the built environment on the NCD unilaterally, and there are few studies that combine the two together. Concurrently, there also exist some problems, such as, the data sources that fail to be updated in time, the unreasonable questionnaire design and some individual data with less explanatory power. Based on this, taking account of the incompleteness of the existing research, this paper selects Xigu District, Lanzhou City, a heavy industry region, as the research area and collects the NCD outpatient data from Xigu District Hospital as the research object to make its spatio-temporal pattern clear. Furthermore, by comparing the data in the heating period with the non-heating period, and by the technology of geo-detector tech, natural environment and built environment factors that affect the NCD are analyzed. Compared with other studies, this paper has the advantage of data acquisition. Meanwhile, the selected heavy industry area can discuss the different research results of the special area and enrich the special case of the existing research, research into the built environment and natural environmental factors of chronic diseases can provide references for the prevention and control of NCD and the improvement of the health status of local residents.

## 2. Materials and Methods

### 2.1. Overview of the Study Area

Located at 36 degrees 05 min 18 s north latitude and at longitude of 103 degrees 37 min 40.70 s east, Xigu District of Lanzhou City covers a total area of 385 km$^2$ with a population of 369,000 (2019), and there are 5 towns, 1 township, 40 administrative villages and 8 subdistricts and 70 communities under its jurisdiction. As a core manufacturing area in Lanzhou City and the largest petrochemical base in west China, it has formed an industrial system propped up with three sectors: petrochemicals, energy, equipment manufacturing and new materials. Most large state-owned enterprises in Xigu were founded during the "First Five-Year Plan" and "Three-line" construction periods which implied a special economic and national defense background; as a result, a typical job-residential integration pattern was formed. Due to an adjoining location between the industrial zone and dwellings (Figure 1). The paper selects the central downtown where distributes dense heavy chemical industry in Xigu District as the study area which includes: Lintao Street, Xigucheng Street, Sijiqing Street, Fulilu Street, Xianfenglu Street, Chenping Street, and Xiliugou Street.

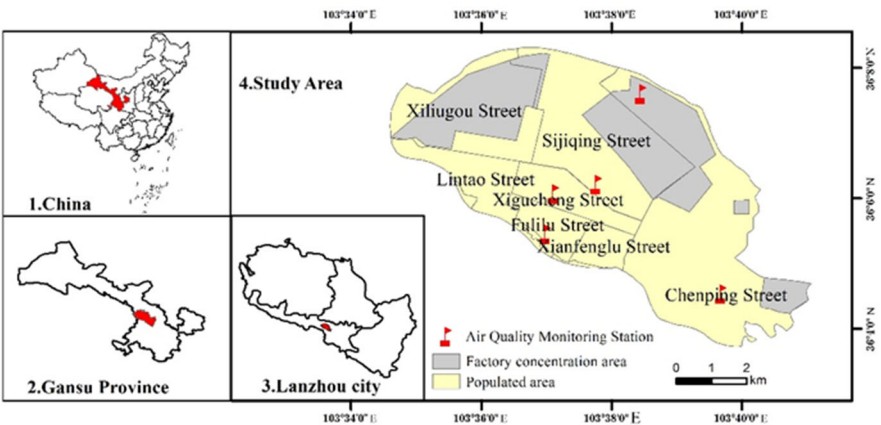

**Figure 1.** Overview of the study area.

### 2.2. Data Sources

The data of NCD comes from in-patients of a hospital in Xigu District, Lanzhou City from 2012 to 2019 and there are 15,819 in total covering: hypertension (1388 cases), diabetes (3007 cases), chronic obstructive pneumonia (7556 cases), heart disease (683 cases), and others (3185 cases). The above data is only used for this study and patients' privacy has been handled properly. POI data (2012–2019), obtained after crawling and cleaning through Baidu Point of Interest websites, is filtered based on the actual situation in Xigu District, and five types of built environment are picked out: greening, medical treatment, factory, transportation, and catering after unwanted data are removed (Table 1). Air quality data (2012–2019) is sourced from China National Environmental Monitoring Center (http://www.cnemc.cn/ accessed date: 10 December 2020). As the study area has the characteristics of a heavy industry area, three indicators are used for description: PM2.5 is selected to represent the concentration of fine particles, and $SO_2$ and $NO_2$ are used to reflect the pollution level of industrial emissions.

**Table 1.** Statistics of natural environment and built environment factors in Xigu district of Lanzhou City.

| Factors | Top Order | Secondary Order | Expected Direction | Amount |
|---|---|---|---|---|
| Air pollution | $SO_2$ | | Positive | 2920 |
| | PM2.5 | | Positive | 2920 |
| | $NO_2$ | | Positive | 2920 |
| Greening rate | Greening facilities | Road greening, greenbelts in neighborhood, parks, etc. | Negative | 316 |
| Medical resources | Medical facilities | Hospitals, clinics, health centers, etc. | Negative | 1533 |
| Industrialization level | Factory facilities | Chemical plants, oil refineries, mechanical processing plants, cotton mills, power plants, etc. | Positive | 840 |
| Traffic network | Traffic facilities | Bus stations, train stations, toll stations, parking lots, etc. | Positive | 3542 |
| Dietary habits | Catering service | Chinese restaurants, foreign-style restaurants, casual dining restaurants, coffee houses, etc. | Positive | 3725 |

### 2.3. Research Methods

2.3.1. Moran Index

Moran Index is applied in this paper to evaluate the spatial autocorrelation of NCD in Xigu District. Its ultimate principle runs as follows:

$$I = \frac{n \sum_{i=1}^{n} \sum_{j=1}^{n} w_{i,j} z_i z_j}{s_0 \sum_{i=1}^{n} z_i^2} \tag{1}$$

where $I$ is Moran's $I$ index, and the value of $I$ goes between $-1$–$1$. A positive value is a positive correlation, and vice versa. The positive correlation is positive, and vice versa. The higher the value, the stronger the spatial autocorrelation of NCD incidence. $Z_i$ refers to the deviation of the attribute of the element i from the average value $(X_i - \overline{X})$, and $W_{i,j}$ means the spatial weight between the element $i$ and $j$. $n$ is equal to the total number of elements, and $S_0$ is the aggregation of all the spatial weights.

$$s_0 = \sum_{i=1}^{n} \sum_{j=1}^{n} w_{i,j} \tag{2}$$

$$E(I) = \frac{-1}{n+1} \tag{3}$$

And when Moran's $I < E(I)$, it means negative spatial correlation while Moran's $I > E(I)$, it means positive spatial correlation, and when Moran's $I = E(I)$, it means zero spatial correlation. Firstly, we created a fishnet of $30 \times 30$ in Xigu District, then counted the number of NCD in each fishnet, and analyzed the spatial autocorrelation of NCD in Xigu District with a single fishnet as the smallest unit. This method not only meets the basic data requirements of Moran index analysis but conforms to the first law of geography as well. It can take into account the continuity of space, and what's more, can be in line with the spatial distribution characteristics of geographical things.

2.3.2. Geographic Detector

The paper adopts the interaction detection in the geographic detector, which is commonly used to detect the spatial distribution of $Y$ and to find how much the detection factor $X$ explains the spatial distribution of the attribute $Y$. By comparing the values of $X1$ and $X2$ with the interacted value, whether the two independent variables work together on the dependent variable $Y$ can be judged.

$$q = 1 - \frac{\sum_{h=1}^{n} N_h \sigma_h^2}{N \sigma^2} = 1 - \frac{SSW}{SST} \tag{4}$$

$$SSW = \sum_{h=1}^{L} N_h \sigma_h^2, SST = N \sigma^2 \tag{5}$$

where the value of $q$ measures the spatial distribution, the value range of $q$ goes between [0, 1]; As $h = 1, 2..., L$ is the stratification of variable $Y$ or the factor $X$; In the formula, $N_h$ and $N$ refer to the number of $h$ and that of units in the whole area, respectively; $\sigma_h^2$ and $\sigma^2$ refer to the variance of $h$ and that of $Y$ in the whole area, respectively; $SSW$ and $SST$ are the sums of the intra-stratal variance and the total variance of the whole area separately.

It can be seen from Table 2 that the eight probe factors selected in this paper are composed of natural environment and built environment factors. The value of $q$ calculated by the geographic detector is used to evaluate the influence of this factor on the NCD. That is to say, the larger the value is, the higher the influence of the factor has, and vice versa.

**Table 2.** Statistics of probe factors.

| No. | Probe Factors | Indexes |
| --- | --- | --- |
| X1 | $SO_2$ | Industrial emission concentration |
| X2 | PM2.5 | Concentration of fine particles |
| X3 | $NO_2$ | Industrial emission concentration |
| X4 | Catering facilities | Amount |
| X5 | Factory facilities | Amount |
| X6 | Traffic facilities | Amount |
| X7 | Greening facilities | Amount |
| X8 | Medical facilities | Amount |

## 3. Results

### 3.1. Significant Change Characters in the Time Series of NCD in Xigu District

It can be seen from Figure 2 that the incidence rates of NCD from 2012 to 2019 were 13.03%, 12.37%, 9.64%, 10.95%, 11.85%, 13.70%, 14.85%, and 13.67%. In terms of years, the rate of NCD presented a trend with fluctuation within the given time from 2012 to 2019 and the average incidence rate reached 12.51%. In view of the actual situation, Lanzhou provides heating on a regular basis every year from 1 November to 31 March of the following year. By comparing the average rates of NCD between the heating period from 2012–2019 and the non-heating period, it is found that the number of patients during the heating period (spring and winter season) was relatively high, accounting for 50.01% of the total patients; while non-heating period (summer and autumn season) witnessed a lower rate making up 49.99% in all the patients. The average incidence of NCD in each month during the heating period increased by 41.86% compared to that of in the non-heating period. In terms of months, the incidence of NCD fluctuated greatly in a year. December was the month with the highest average value of NCD having an incidence rate of 1.30% and accounting for 10.37% of the annual average; while June was the lowest one with an incidence rate of 0.83%, accounting for 6.65% of the annual average.

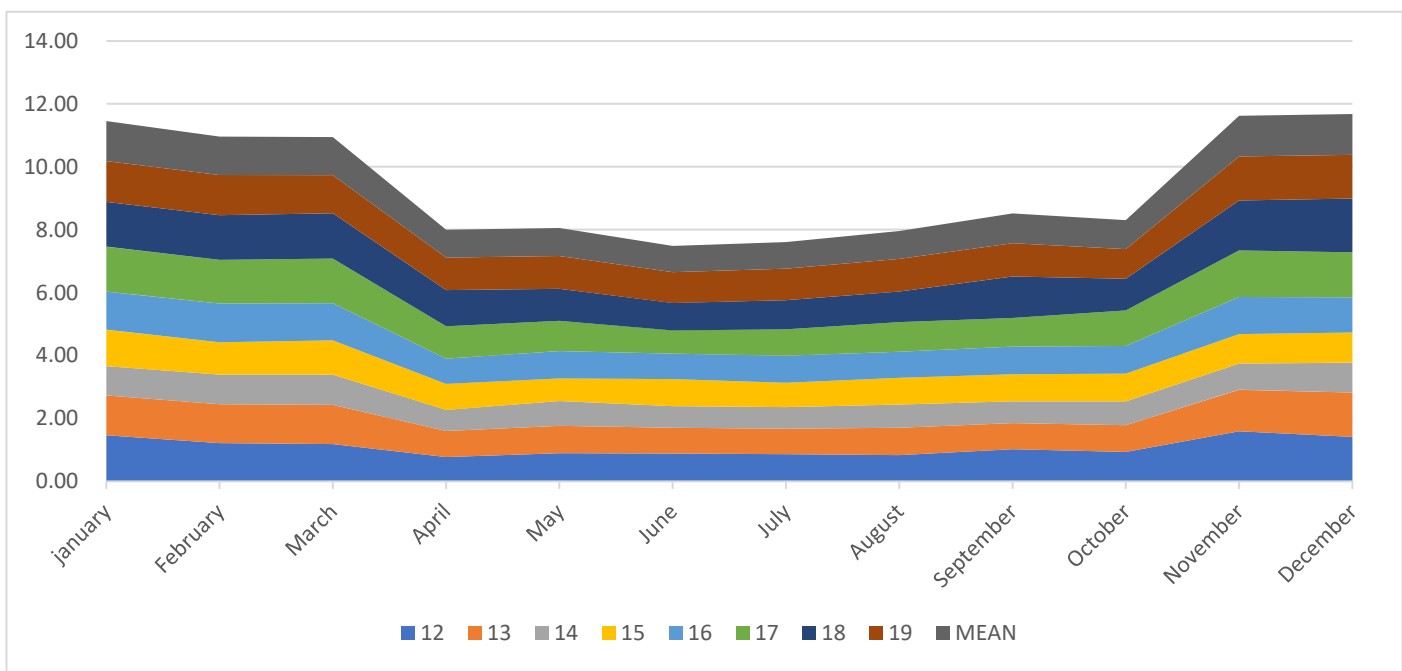

**Figure 2.** NCD incidence statistics from 2012 to 2019 (%).

Figure 3 depicts the time series characters between NCD and the monthly average concentrations of PM2.5, of $SO_2$ and of $NO_2$ from 2012 to 2019. The specific changes are

shown as follows: the peak number of NCD patients and the peak concentration values of the three variables were all fastened on the heating period in a year (spring and winter season) rather than the non-heating period (summer and autumn season, and October) in which each variable had a decline to a varying degree. The number of NCD patients in the non-heating period was witnessed a larger fall than that of in the heating period, accompanying an overall large fluctuation, with a variance of 1356.59. In the meantime, the concentration of the other three variables in the non-heating period and the heating period experienced a small decrease with a slight overall fluctuation. The variances of the PM2.5 concentration, $SO_2$ concentration and $NO_2$ concentration were 249.69, 111.59 and 128.01, respectively. When the concentrations of PM2.5, $SO_2$ and $NO_2$ rose, the number of patients with NCD went up accordingly, and vice versa indicating that there might be a positive correlation between NCD and these three variables.

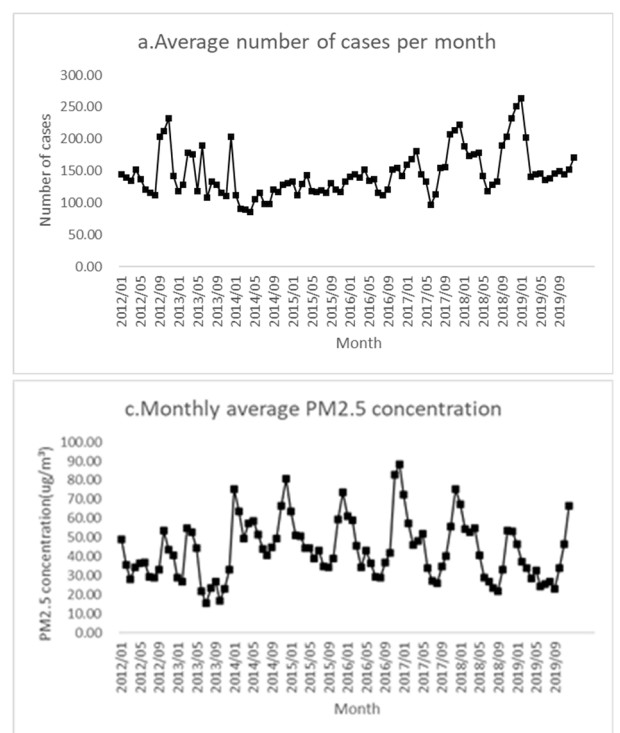
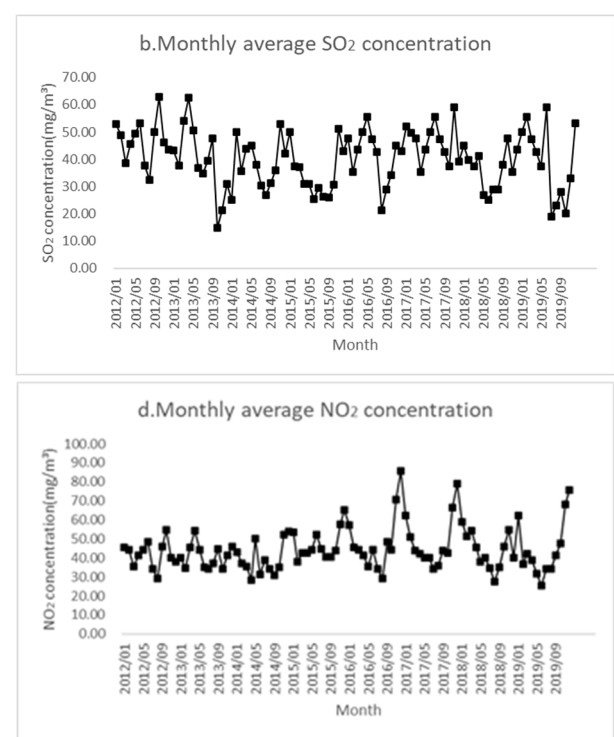

**Figure 3.** Time series analysis of natural environmental factors and NCD.

### 3.2. Spatial Distribution Characteristics of NCD in Xigu District

The spatial distribution of NCD over time is illustrated in Figure 4, The average values of NCD incidence from 2012 to 2019 revealed that NCD in Xigu District showed a decreasing trend from the high incidence in the central area to the surrounding areas, which was shown as follows: Xigucheng Street (22.71%) > Fulilu Street (18.46%) > Sijiqing Street (14.47%) > Xianfenglu Street (13.26%) > Lintao Street (12.05%) > Chenping Street (10.715%) > Xiliugou Street (8.34%).

Taken Figure 4 and Table 3 for reference, in terms of time, the incidence in Lintao street increased by 39.70% in 2012 and 2013 while the changes in other streets were relatively small. In 2014 and 2015, the incidence in Sijiqing Street decreased by 68.10%, while the incidence in Chenping Street and Fulilu Street showed an upward trend, increasing by 5.89% and 23.32%, respectively. In 2016 and 2017, the NCD high-incidence areas gradually shifted from the east of Xigu District to the north. The incidence of Sijiqing Street soared sharply (102.54%), while the incidence in Xianfenglu decreased by 28.57%, and the rest of the streets had a small drop. In 2018 and 2019, the incidence of Xiliugou Street increased by 116.03%, while the incidence of Sijiqing Street decreased by 17.93%, and the remaining streets changed little. In terms of all these streets, from 2012 to 2019, Xigucheng Street

had always been a high-risk area with an average incidence rate of 22.71%, much higher than others. Xiliugou Street, however, was a low incidence area for NCD, with an average incidence of 8.34%.

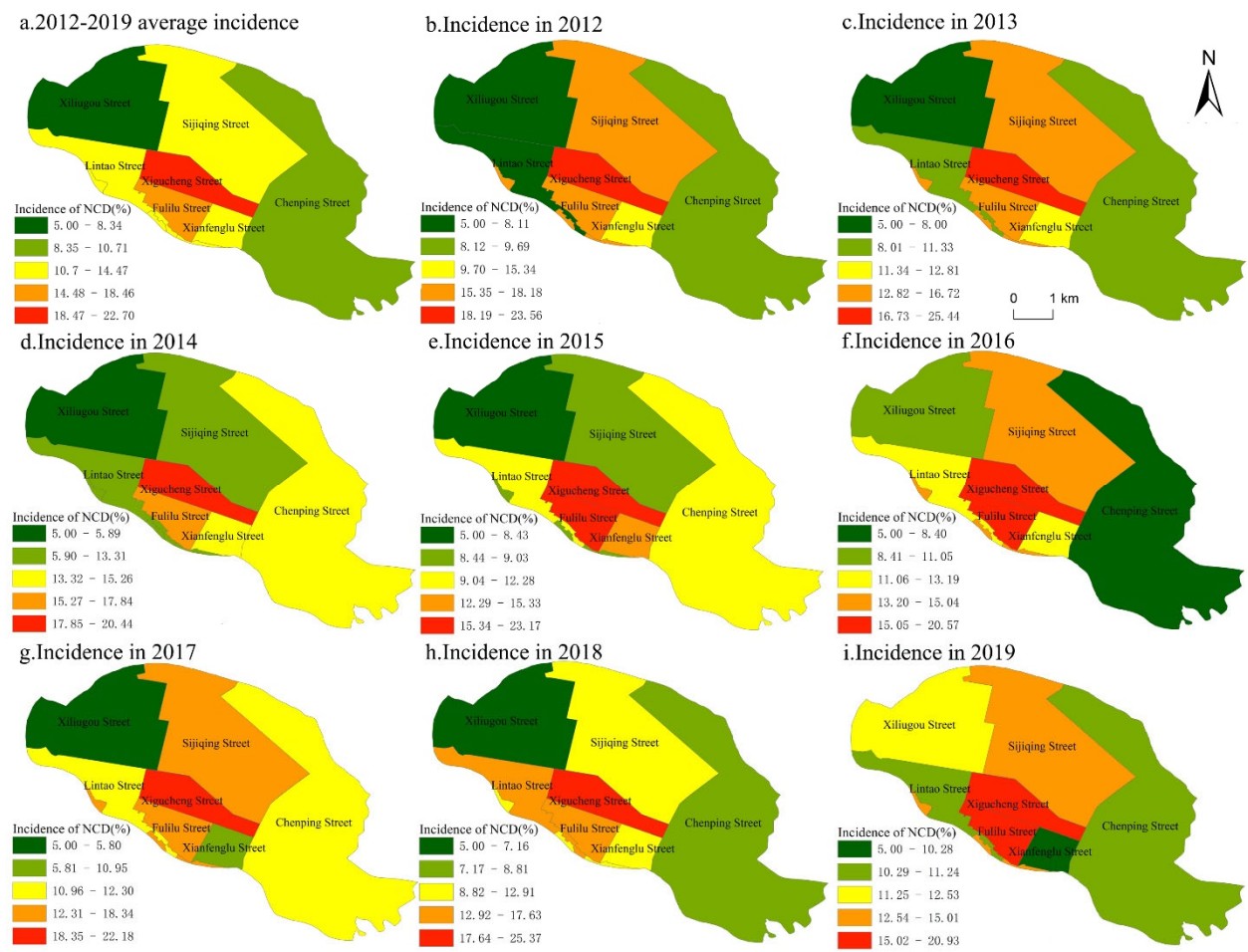

**Figure 4.** NCD incidence distribution in Xigu district, 2012–2019.

**Table 3.** Annual variation of NCD prevalence in streets of Xigu District, 2012–2019(%).

| Year | 2012 | 2013 | 2014 | 2015 | 2016 | 2017 | 2018 | 2019 | Average Value |
|---|---|---|---|---|---|---|---|---|---|
| Xigucheng Street | 0.85 | 2.73 | −2.27 | 0.46 | −2.14 | −0.53 | 2.66 | −1.78 | 22.71 |
| Fulilu Street | −0.28 | −1.74 | −0.62 | 2.16 | 0.65 | −0.12 | −0.83 | 0.79 | 18.46 |
| Xianfenglu Street | 2.08 | −0.45 | 2 | 2.07 | −0.07 | −2.31 | −0.35 | −2.98 | 13.26 |
| Lintaojie Street | −3.94 | −0.72 | 1.13 | 0.23 | 0.59 | 0.09 | 3.39 | −0.81 | 12.05 |
| Sijiqing Street | 2.78 | 0.71 | −1.16 | −5.44 | 0.57 | 3.82 | −1.79 | 0.54 | 14.47 |
| Chenping Street | −1.02 | −0.19 | 3.37 | 0.43 | −2.31 | 1.59 | −1.9 | 0.05 | 10.71 |
| Xiliugou Street | −0.47 | −0.34 | −2.45 | 0.09 | 2.71 | −2.54 | −1.18 | 4.19 | 8.34 |

*3.3. Obvious High Cluster of NCD during the Heating Period and the Non-Heating Period*

Taking the incidence of NCD in Xigu District in 2019 as an example, the paper used the local Moran index to study the spatial heterogeneity of NCD in each month, as shown in Figure 4: during the heating period (spring and winter season), there were a large number of high-high cluster areas (Figure 5a–c,k,l), mainly concentrated in the Center of Xigu District. In the non-heating period (summer and autumn season), high-high cluster area was in the Center of Xigu District (Figure 5d–j). This finding shows that: the spatial distribution of NCD in the Center of Xigu District changed accordingly due

to heating, and the number of NCD cases in the streets adjacent to the industrial zone increased significantly during the heating period while that of in the streets farther from the industrial zone did not vary greatly; The spatial distribution changes of NCD in these streets during the non-heating period were relatively small. However, The Center of Xigu District was always the area with high incidence of NCD before or after the heating period.

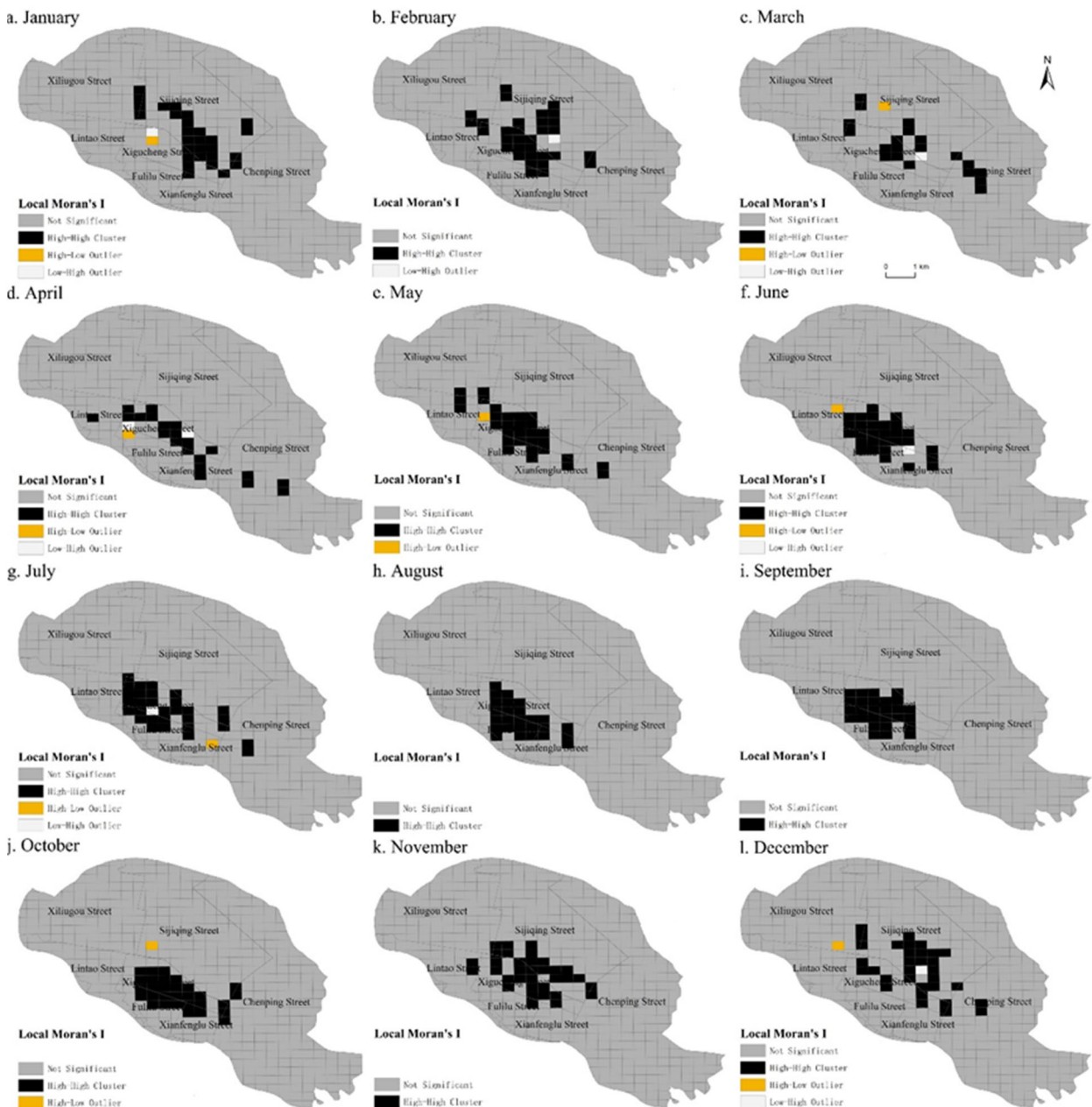

**Figure 5.** NCD mean local spatial autocorrelation for each month of 2019.

### 3.4. Natural Environment and Built Environment Factors Applied Significantly Different Acting Force on NCD

According to the results of the geo-detector (Table 4), the acting force of each factor on the NCD distribution went downwards all the way such as this: factory facilities (q = 0.7483) > $SO_2$ (q = 0.7230) > $NO_2$ (q = 0.6993) > PM2.5 (q = 0.6845) > catering services (q = 0.6620) > greening facilities (q = 0.6420) > traffic facilities (q = 0.5797) > medical facilities (q = 0.2744).

**Table 4.** Differentiation and factor detection results.

|   | SO$_2$ | PM2.5 | NO$_2$ | Catering Service | Factory Facilities | Traffic Facilities | Greening Facilities | Medical Facilities |
|---|--------|-------|--------|------------------|--------------------|--------------------|---------------------|--------------------|
| q | 0.7230 | 0.6845 | 0.6993 | 0.6620 | 0.7483 | 0.5797 | 0.6420 | 0.2744 |
| *p* | 0.0000 | 0.0000 | 0.0000 | 0.0000 | 0.0000 | 0.0000 | 0.0000 | 0.0000 |

In the natural environment, the concentration of PM2.5 and SO$_2$ had a heavy toll on the incidence of NCD, and the three factors all presented a positive correlation with the incidence of NCD. Air pollutants themselves, as the inducing factors of NCD, affected the incidence of NCD. Taking the characteristics of Xigu District into consideration, industrial production emitted a large amount of air pollutants, resulting in the air quality level in heavy industry areas being far worse than other areas, which in turn, further deepened the impact of natural environmental factors on the incidence of NCD.

In the built environment, the factory facilities had the strongest explanatory power of NCD. Xigu District, as a heavy industry base, hosts a large number of petrochemical companies whose industrial production brings negative sides to the environment, resulting in the number and distance of factory facilities greatly affect the incidence of NCD in Xigu District. In view of this, Xigu District should reasonably program the location of residential areas and control the distance between factory facilities and residential areas to alleviate the impact of factories on the incidence of NCD in the area. The explanatory power of catering services for NCD shows that: the growth of catering service facilities would affect the diet structure and eating habits of surrounding residents to a certain extent, and then would affect the incidence of hypertension, diabetes and other diseases; that is to say, the distribution of catering service facilities had a positive correlation with NCD in Xigu District. However, that as it may, it is an indispensable existence that catering services are. Only when residents adjust their own eating habits can the impact of catering services on NCD be fundamentally lessened. The explanatory power of transportation facilities for NCD indicated that the incidence of NCD in residential areas with dense traffic facilities was showing an upward trend. Greening facilities' explanation for NCD told us that the higher the greening rate in residential areas was, the lower the incidence of NCD had. In other words, the incidence of NCD can be effectively reduced by increasing the number of greening facilities around residential areas. As for the explanatory power of medical facilities on NCD, it showed that the number of medical facilities had a slight influence on the incidence of NCD. Chronic diseases are seen as non-communicable diseases. There is little effect in reducing the incidence of NCD by increasing the number of medical facilities, which again confirms the fact that prevention and control is the most effective way for the reduction of the incidence of NCD. From now on, we should publicize more about the prevention and control of chronic diseases, raise residents' awareness and improve the natural environment and built environment, which as a whole can achieve the goal of reducing the incidence of NCD.

It can be seen from Figures 6 and 7 that the interaction between factory facilities and SO$_2$ had the strongest explanatory power (q = 0.98862), which almost determined the distribution of NCD. The explanatory power of the interaction was greater than the explanatory power of single factor on NCD, for the type was classified as the two-factor enhanced one. The number of factories in the streets of Xigu District was directly proportional to the incidence of NCD. Factory facilities, as a static built environment factor, exerted an impact on the health of surrounding residents with the characteristics of long duration and small changes in the effecting degree, especially in Xigu District where such features were more obvious under the job-residential integration pattern. Combining the conclusions obtained in Figure 3, the impact of air pollutants (being viewed as a dynamically changing natural environmental factor) on the residents' health varied with the time sequence, which verified the hypothesis proposed in this paper that the natural environmental factors and built environmental factors acted on NCD together. Among them, air pollutants were seen as the direct path of influence, while factory facilities as the indirect path. That is to

say, the emission of air pollutants from the factory facilities led to a decline in air quality, which affected the incidence of NCD in Xigu District. All types of factors interacting ones belonged to the two-factor enhanced types whose explanatory power was larger than the power by a single factor, indicating that the pathogenic process of NCD was a comprehensive process, determined by the interaction between the natural environment and the built environment factors.

|  | SO$_2$ | PM2.5 | NO$_2$ | Catering services | Factory facilities | Traffic facilities | Greening facilities |
|---|---|---|---|---|---|---|---|
| PM2.5 | 0.79421 |  |  |  |  |  |  |
| NO$_2$ | 0.78213 | 0.71059 |  |  |  |  |  |
| Catering services | 0.93995 | 0.93547 | 0.93988 |  |  |  |  |
| Factory facilities | 0.98862 | 0.96941 | 0.96773 | 0.94358 |  |  |  |
| Traffic facilities | 0.92908 | 0.91013 | 0.93997 | 0.95038 | 0.92745 |  |  |
| Greening facilities | 0.9289 | 0.91949 | 0.90949 | 0.91953 | 0.95385 | 0.68222 |  |
| Medical facilities | 0.61104 | 0.56633 | 0.5723 | 0.56173 | 0.59255 | 0.6888 | 0.59459 |

**Figure 6.** Interaction detector results.

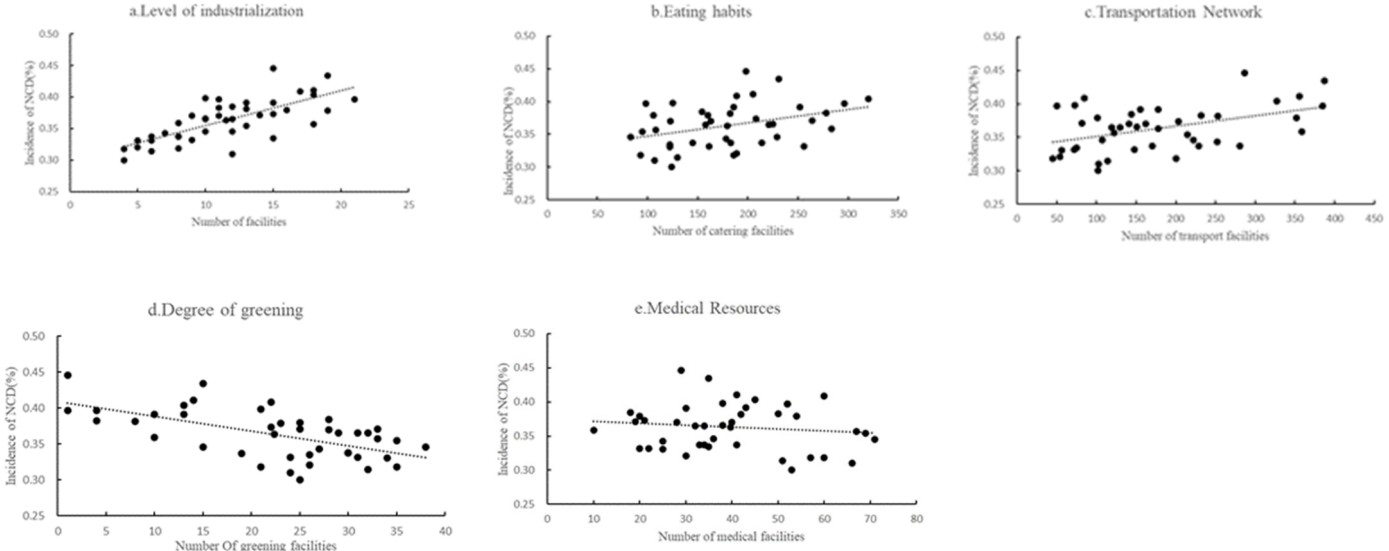

**Figure 7.** Correlation analysis between environmental factors and incidence of NCD.

## 4. Discussion and Outlook

### 4.1. Main Findings

Some scholars found time series changes between ambient air pollution and hypertension [35–37]. T The study found that the incidence of chronic non-communicable disease fluctuates over time and increases slowly. The conclusions of this study are similar to those of other cross-sectional studies or group studies. Most of the existing studies discuss the seasonal characteristics of the incidence of NCD, the research has found that the incidence in summer and autumn is higher than that in spring and winter [38–41]. But this paper, based on the characteristics of the study area, compares the incidence of NCD during the heating period and the non-heating period for the first time, and finds that the incidence of NCD during the heating period was greater than that in the non-heating period, As a control group before and after heating period, the study of the difference in the incidence of chronic Non-communicable disease is closer to the actual life of residents, which can be seen that there are variations in the incidence of NCD in different study areas.

Mainstream research has discussed the relationship between infrastructure, Transportation Planning, air pollutants and green space, and access to health care resources and the incidence of chronic diseases such as hypertension, heart disease and Obstructive lung disease [42–47]. These studies have shown that chronic diseases are affected by many factors and have shown that chronic diseases are affected by a single factor, but little has been carried out about the interaction of these two or more factors with NCD. This study examines the interaction of the natural and built environment with NCD, it is found that $SO_2$, PM2.5, $NO_2$, diet, traffic, greening and medical treatment in natural and built environment are the key factors affecting NCD in Xigu district, the incidence and spatial distribution of chronic non-communicable disease are almost entirely determined by factory facilities and air quality. The reason for this difference lies in the characteristics of heavy industry in the study area. The advantage is that it can well represent the heavy industry area, but the disadvantage lies in the lack of control group experiments, which cannot be well applied to other areas.

### 4.2. Strengths and Limitations

Selecting Xigu District of Lanzhou City as the research area, based on its own characteristics as a heavy industrial area, the paper takes fully consideration of the interaction between natural environment and built environment factors on NCD, but we've to say there are still a few weaknesses: (1). This paper collects 2012–2019 annual data in Xigu District on various types of chronic diseases. However, chronic diseases actually cover a variety of diseases whose characteristics and impact factors are not the same. In our research, all these diseases are generalized as chronic diseases to explore the relationship between NCD and the natural environment and built environment. Whether the results are convincing enough still needs further verification, and the POI data does not fully represent the built environment factors. In future studies, more comprehensive data will continue to be collected for a strong support. (2). The terrain of Xigu District is complex, and the distribution of residential areas and population activities are relatively clustered. This paper takes this area as an example to describe the temporal and spatial distribution of NCD in all heavy industrial areas and its influencing factors, which obviously lacks explanation for the distribution of NCD in other cities with different location characteristics. Since the research cannot be compared with other areas, it appears to be one-sided, leading to a narrow range of detection results. In the future, more comparisons with other research areas should be made to improve the accuracy of the results and the scope of application also can be expanded. (3). This paper only studies the impact of natural environment and built environment factors on chronic diseases and does not consider factors such as social environment and individual behavior and fails to take the active choices of individual behavior into consideration. In future, more impact factors should be supplemented into the research to improve the scientificity and accuracy of this paper.

### 4.3. Recommendations and Prospects

In summary, the spatio-temporal distribution of NCD presents obvious characteristics, and there are evident differences before and after the heating duration. The impact of various natural environment and built environment factors on NCD is varied. It is properly meaningful to start with improving the natural environment and built environment as a way of controlling and preventing NCD. Under the unique job-residential integration pattern of Xigu District, the service facilities around the residential areas should be reasonably configured, such as controlling the number of factory facilities around the residential areas, increasing the number of greening facilities, and considering the establishment of detecting points for air quality in industrial areas and residential areas. In this way, air pollutants will be detected in real time, and quantitative evaluation standards also will be established to reduce industrial emissions. Furthermore, for the sake of disease prevention and control, the publicity work on the prevention of NCD shall be strengthened, which will help reveal the main causes of disease (industrial facilities, air pollution), and develop

a healthy lifestyle. In doing so, the incidence of NCD will be dropped and the health of residents will be ensured as well.

## 5. Conclusions

By collecting data on NCD patients in Xigu District, Lanzhou City from 2012 to 2019, this paper applies statistical methods and local Moran index into use to analyze the time sequence and spatial distribution of NCD. What's more, by comparing the space-time characteristics during the heating period and during the non-heating period with consideration of the actual situation, and by using the geographic detector to study the relationship between NCD and natural environment and built environment factors, the paper sums up following conclusions:

NCD in Xigu District presented obvious temporal and spatial characteristics. In terms of time, the incidence of NCD in Xigu District from 2012 to 2019 fluctuated over time. In terms of space, the NCD in Xigu District generally showed a downward trend from the high incidence in the central area to the surroundings. There was a clear contrast between the heating period and the non-heating period. In terms of time, the incidence of NCD was greatly affected by the heating period. In the heating period (spring and winter season), the incidence was higher than that of the non-heating period (summer and autumn season). In terms of space, there were quite a few H-H cluster areas during the heating period, mostly locating in Xigucheng Street and Sijiqing Street while H-H areas in the non-heating period were mainly concentrated in Xigucheng Street.

Natural environment and built environment factors had different paths of action on NCD. The explanatory power of the interaction between natural environment and built environment factors was stronger than that of any single one. All the interactions among impact factors fell into a two-factor enhanced relationship. The paths of the two factors on NCD went similar to this: Path 1, the natural environment and built environment directly affected the incidence of NCD. Path 2: The built environment treated the natural environment as an intermediary variable to indirectly affect NCD. Industrial production in factories caused an increase in the concentration of air pollutants, which not only affected air quality but triggered changes in the incidence of NCD.

**Author Contributions:** Conceptualization, H.Z. and Y.D.; methodology, J.L.; software, M.W.; validation, Y.D., F.Z. and M.W.; formal analysis, J.L.; investigation, H.Z.; resources, H.Z.; data curation, Y.D.; writing—original draft preparation, Y.D.; writing—review and editing, H.Z. and Y.D.; visualization, Y.D.; supervision, H.Z.; project administration, H.Z.; funding acquisition, Y.D. All authors have read and agreed to the published version of the manuscript.

**Funding:** This research was funded by THE PROJECT OF NATIONAL SOCIAL SCIENCE FOUNDATION OF CHINA (NO. 41971268).

**Institutional Review Board Statement:** Not applicable.

**Informed Consent Statement:** Not applicable.

**Data Availability Statement:** Not applicable.

**Conflicts of Interest:** The authors declare no conflict of interest.

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
