# Peer review of "The Influence of Environment Factors on Chronic Non-Communicable Diseases in a Heavy Industry City—A Case of Xigu District of Lanzhou City"

_sustainability, doi:10.3390/su132212636_

Round 1
Reviewer 1 Report
The long time scale of this study together with its focus on non-lifestyle factors’ roles in chronic diseases make this a valuable contribution to the literature. However, I did find the paper difficult to read in some sections so the manuscript would benefit from editing by an English speaking editor.
Specific comments below:
Abstract
Line 10 – remove ”based on the statistical method and Moran Index”
Define H-H
Do not refer to specific streets in the abstract – these are meaningless for those not familiar with the area
Rewrite the abstract to indicate what was found, not just that the variables all affect the incidence to different degrees - for example factory facilities and SO2 had the strongest link with NCD
Please change “heating period” to season and indicate the seasons – summer, winter, etc. do this throughout the manuscript.
Introduction
Line 32 – remove “on earth”
Lines 36-38 – this doesn’t make sense – how will the fatalities caused by chronic disease be reduced by 30%? Is that a target the of the plan? If so that needs to be clear.
Lines 57 -58 – “easily induce disease such as high blood pressure…” remove this sentence or add a number of citations to support it.
Line 91-92 – can the authors say this in the introduction>? Is this not what the study was looking for?
Methods
Page 4 – please add an explanation of what each index or test aims to find
Results
Table 3 is meaningless to a reader. either graph this information or collate it so that the reader can make sense of what is being presented
Discussion
Much of the discussion is a repeat of the results. It would be better to discuss those results with reference to the literature – are they expected? How does this compare with other studies?
Author Response
Dear reviewer, after a careful reading of your comments, we have made the following revisions,Please see the attachment,Thank you very much for your comments, is my thesis has been greatly improved, I wish you a happy life, work smoothly.

Reviewer 2 Report
My modest advice is to check the syntax and word choices of the entire manuscript to ensure that the intended meaning is conveyed. For example, in Line 18, “was witnessed” should be “has witnessed”. In Line 41, “the influencing factors of” should be “the factors influencing”, etc.
Introductory section: A more robust case should be made regarding the significance of the present study. Namely, explain more clearly and with more specificity the contribution that the study makes to the extant literature.
Method section: Explain more clearly the rationale for area selection. This information may then be used to contextualize the findings of the present study in the discussion section. For instance, in the discussion section, more specificity needs to be given to how the findings of the selected area compare with the findings of other areas that were not selected.
Result section: The evidence provided by the authors is mainly descriptive. Is there a way to introduce inferential statistics so that differences and trends can be considered statistically significant and/or the relative contribution of each factor can be precisely measured (e.g., regression analysis)?
Discussion section: Please re-write this section to ensure clarity. Subsections can be created to organize topics. One subsection may be focused on summarizing the key results of the study and describing how they compare with existing ones. Another subsection may deal with potential remedies based on current and past evidence.
Line 311. Please clarify the following sentence: “Chronic diseases are affected by a variety of impact factors, and the mechanism of each factor is differentiated.”. It is unclear and not well linked to the sentence that comes after it.
Please rephrase the sentence that contains “for existing researches believe” (Line 319). You may consider noting that researchers have uncovered evidence supporting a particular viewpoint.
Author Response

(The authors gave the same response as above.)

Round 2
Reviewer 2 Report
The paper has considerably improved in overall quality. My modest advice is to further proofread the current text for grammatical structure and word choices.
This manuscript is a resubmission of an earlier submission. The following is a list of the peer review reports and author responses from that submission.